# Clinical features associated with Neo*RAS* wild-type metastatic colorectal cancer A SCRUM-Japan GOZILA substudy

Hiroki Osumi [1], Eiji Shinozaki [1] ✉, Yoshiaki Nakamura [2], Taito Esaki[3], Hisateru Yasui [4], Hiroya Taniguchi[5], Hironaga Satake [6], Yu Sunakawa [7], Yoshito Komatsu [8], Yoshinori Kagawa[9], Tadamichi Denda[10], Manabu Shiozawa[11], Taroh Satoh[12], Tomohiro Nishina[13], Masahiro Goto[14], Naoki Takahashi[15], Takeshi Kato[16], Hideaki Bando [2], Kensei Yamaguchi[1] & Takayuki Yoshino [2] ✉

"Neo*RAS* WT" refers to the loss of *RAS* mutations (MTs) following first-line treatment in metastatic colorectal cancer (mCRC). We evaluate the incidence and clinicopathological characteristics of Neo*RAS* WT mCRC using next-generation sequencing of plasma circulating tumor DNA. Patients with mCRC enrolled in the GOZILA study initially diagnosed with tissue *RAS* MT mCRC and received subsequent systemic therapy are eligible. Neo*RAS* WT is defined as the absence of detectable *RAS* MT in plasma and assessed in all eligible patients (Group A) and in a subgroup with at least one somatic alteration detected in plasma (Group B). Overall, 478 patients are included. Neo*RAS* WT prevalence is 19.0% (91/478) in Group A and 9.8% (42/429) in Group B. Absence of liver or lymph node metastasis and tissue *RAS* MTs other than *KRAS* exon 2 MTs are significantly associated with Neo*RAS* WT emergence. Overall, 1/6 and 2/6 patients with Neo*RAS* WT treated with anti-EGFR monoclonal antibodies (mAbs) show partial response and stable disease for ≥6 months, respectively. Neo*RAS* WT mCRC is observed at a meaningful prevalence, and anti-EGFR mAb-based therapy may be effective.

*RAS* genes (*KRAS, NRAS,* and *HRAS*) are oncogenes that produce RAS proteins responsible for transmitting signals that promote cell growth[1]. *RAS* mutations (MTs) are associated with carcinogenesis in many cancers; however, effective therapies for these cancers are limited[2]. Overall, 17% of solid tumors involve *KRAS* MTs, including ~50% of metastatic colorectal cancer (mCRC) cases[3,4]. Anti-epidermal growth factor receptor (EGFR) monoclonal antibodies (mAbs), namely, cetuximab and panitumumab, are a mainstay of mCRC therapy, but

[1]Department of Gastroenterological Chemotherapy, Cancer Institute Hospital of Japanese Foundation for Cancer Research, Tokyo, Japan. [2]Department of Gastroenterology and Gastrointestinal Oncology, National Cancer Center Hospital East, Kashiwa, Japan. [3]Department of Gastrointestinal and Medical Oncology, National Hospital Organization Kyushu Cancer Center, Fukuoka, Japan. [4]Department of Medical Oncology, Kobe City Medical Center General Hospital, Kobe, Japan. [5]Department of Clinical Oncology, Aichi Cancer Center Hospital, Nagoya, Japan. [6]Department of Medical Oncology, Kochi Medical School, Kochi, Japan. [7]Department of Clinical Oncology, St. Marianna University School of Medicine, Kawasaki, Japan. [8]Department of Gastroenterology and Hepatology, Hokkaido University Hospital, Sapporo, Japan. [9]Department of Gastroenterological Surgery, Osaka General Medical Center, Osaka, Japan. [10]Division of Gastroenterology, Chiba Cancer Center, Chiba, Japan. [11]Department of Gastroenterological Surgery, Kanagawa Cancer Center, Yokohama, Japan. [12]Palliative and Supportive Care Center, Osaka University Hospital, Suita, Japan. [13]Department of Gastrointestinal Medical Oncology, National Hospital Organization Shikoku Cancer Center, Ehime, Japan. [14]Department of Cancer Chemotherapy Center, Osaka Medical and Pharmaceutical University Hospital, Osaka, Japan. [15]Department of Gastroenterology, Saitama Cancer Center, Saitama, Japan. [16]Department of Surgery, National Hospital Organization Osaka National Hospital, Osaka, Japan. ✉e-mail: eiji.shinozaki@jfcr.or.jp; tyoshino@east.ncc.go.jp

they are ineffective against *RAS* MT mCRC[5,6]. The current clinical practice guidelines recommend *RAS* gene testing before administering anti-EGFR mAb-based therapy to patients with mCRC[7–9]. Given the lack of treatment options for these patients and their poor prognosis[10], further therapeutic developments are urgently needed.

Understanding of the mechanisms by which the *RAS* mutational status evolves before and after treatment has recently advanced, primarily owing to advances in circulating tumor DNA (ctDNA)-based diagnostics that enable minimally invasive, simple, and repeatable testing[11–13]. *RAS* MTs in ctDNA have been identified in patients with anti-EGFR mAb-resistant *RAS* "WT" mCRC, many of whom acquire the condition after exposure to anti-EGFR mAbs[14,15]. Contrarily, initially diagnosed *RAS* MT patients have *RAS* WT after chemotherapy-based treatment. This phenomenon, called "Neo*RAS* WT" mCRC, has attracted research attention because anti-EGFR mAbs, conventionally ineffective against *RAS* MT mCRC, may be effective in Neo*RAS* WT mCRC[16–18]. Several clinical trials on the safety and efficacy of anti-EGFR mAbs for the treatment of Neo*RAS* WT mCRC are ongoing[19,20].

The incidence of Neo*RAS* WT mCRC varied widely from 20% to 80% among previous studies[17,21–25]. However, these studies were limited by small sample sizes, varying lines of treatment, and lack of a consensus definition of the Neo*RAS* WT mCRC population. In particular, when only *RAS* MTs were measured, it was not possible to determine whether *RAS* MTs had disappeared or whether ctDNA was not detectable. Attempts have since been made to assess ctDNA using next-generation sequencing (NGS) or detect methylated ctDNA[21,25].

The SCRUM-Japan GOZILA study is an ongoing Japanese nationwide screening project that analyzes cancer genomic information with NGS from the plasma ctDNA of patients with advanced gastrointestinal cancer using the Guardant360® assay[26]. Initial results demonstrated that ctDNA NGS had a faster turnaround time and greater diagnostic yield than tissue NGS[26]. In addition, other studies have validated ctDNA NGS for the assessment of *RAS* mutations, *BRAF* V600E mutation[27], microsatellite instability[28], and HER2 amplification[29] with high concordance to tissue-based companion diagnostics in patients with mCRC. The present study aimed to use this platform to examine the incidence and clinicopathological characteristics of Neo*RAS* WT mCRC.

## Results

### Patient characteristics

Among the 4991 patients with mCRC enrolled in GOZILA between March 2018 and February 2022, 647 patients were initially diagnosed with *RAS* MT mCRC using pretreatment tissue analysis. Among them, 169 patients in whom ctDNA was measured before the administration of first-line chemotherapy ($N = 152$) and in whom plasma *RAS* MTs were different from the tissue *RAS* MTs ($N = 17$) were excluded. Finally, 478 patients were evaluated. The CONSORT flow diagram is shown in Fig. 1. The median age at the time of blood sampling was 62.0 years (range, 25.0–85.0 years), and 249 (52.1%) patients were male. Among the 478 patients, 153 (32.0%) and 306 (64.0%) patients had right-sided tumors and multi-organ metastatic sites, respectively (Supplementary Data 1). The lungs were the most frequent site of metastasis (60.7%), followed by the liver (56.9%), lymph nodes (28.0%), and peritoneum (27.8%). Furthermore, 357 (74.7%) patients were treated with first-, second-, or third-line chemotherapy at the time of sampling; 129 (27.0%) patients received 5-fluorouracil, capecitabine, or S-1 and irinotecan combination therapy; and 334 (69.9%) patients received anti-vascular endothelial growth factor antibodies before blood collection for ctDNA analysis.

### Association of clinicopathological characteristics with Neo*RAS* WT

Somatic MTs in *KRAS* and *NRAS* exons 2, 3, and 4 were detected in the plasma from 387 of the 478 (81.0%) patients with mCRC. Thus, the incidence of potential Neo*RAS* WT mCRC in Group A was 19.0% (91/478). Among the 91 patients, 30 patients had no detectable genetic alterations, and 2 had putative germline MTs (*BRCA2* and *ATM*). The possibility of clonal hematopoiesis (CH) could not be ruled out in 17 patients (*TP53*, *ATM*, and *GNAS*) without any other somatic alterations. Therefore, the incidence of Neo*RAS* WT mCRC in Group B after excluding these potential confounding factors was 9.8% (42/429). Figure 2 shows the details of other somatic alterations in Group B. The median variant frequency was 0.23% (range: 0.08–29.05), and *APC* mutations were the most frequently detected mutations. When the limits of detection (i.e., minimum detectable mutant allele frequency [MAF]) were defined as 0.34% and 1%, the incidence rates of Neo*RAS* WT mCRC in Group B were 2.5% (10/397) and 1.5% (6/393), respectively.

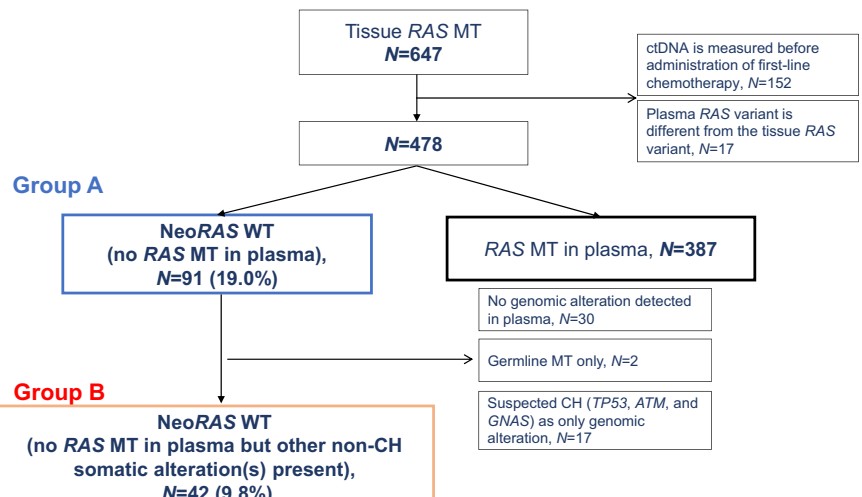

**Fig. 1 | CONSORT diagram of this study.** Tissue samples from 647 *RAS* MT mCRC patients are initially included. Among these, samples from 169 patients are excluded; finally, samples from 478 patients are included in the analysis. Somatic mutations in *KRAS* and *NRAS* are detected in 387 patients with mCRC in their plasma. Thus, the incidence of Neo*RAS* WT mCRC in Group A is 19%. Overall, 30 patients have no detectable genetic alteration, and 2 patients have germline mutations. The possibility of CH cannot be ruled out in 17 patients. Therefore, the incidence of Neo*RAS* WT mCRC in Group B is 9.8%. ctDNA: circulating tumor DNA, *RAS*: rat sarcoma viral oncogene homolog, WT wild-type, MT mutant, CH clonal hematopoiesis, mCRC metastatic colorectal cancer.

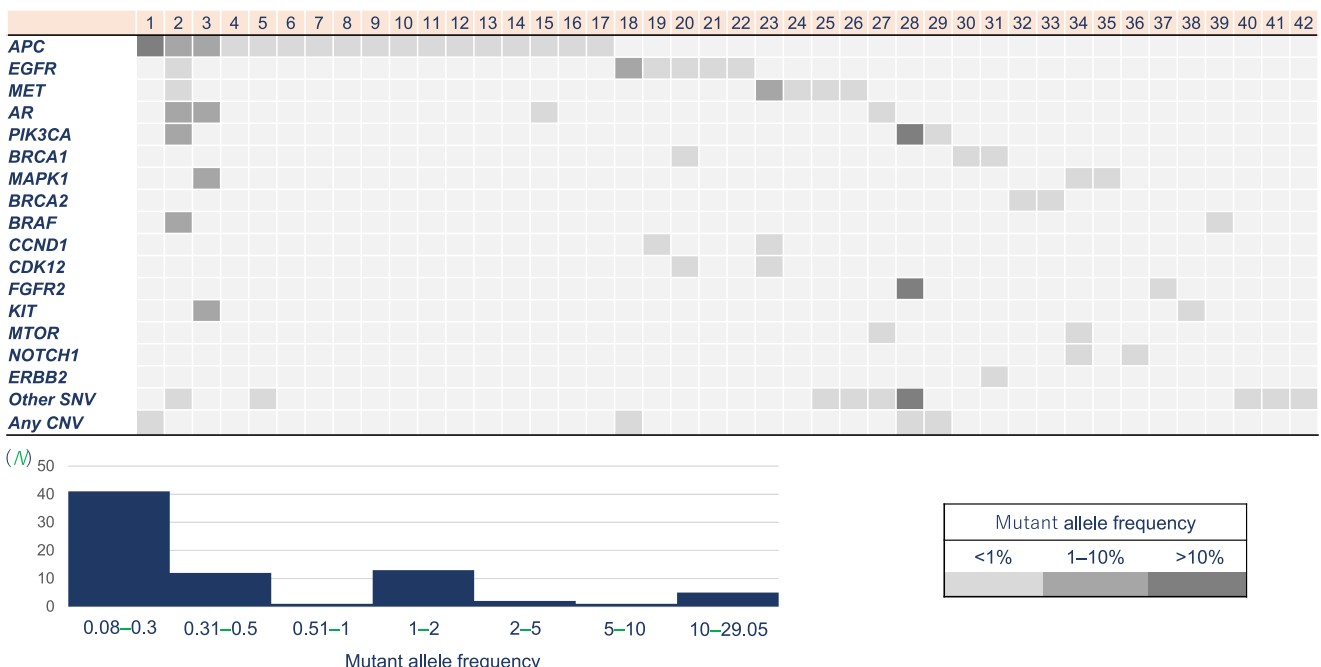

**Fig. 2 | Details of somatic alterations other than *RAS* in Group B.** The median variant allele frequency is 0.23% (range: 0.08−29.05), and *APC* mutations are the most frequently detected. SNV single-nucleotide variant, CNV copy number variant, *RAS* rat sarcoma viral oncogene homolog.

In Group A, there were significant differences in the frequency of Neo*RAS* between single-organ and multi-organ metastatic sites (30.2% vs. 12.7%, *P* < 0.001) and between the absence and presence of liver (34.0% vs. 7.7%, *P* < 0.001), lymph node (22.1% vs. 11.2%, *P* < 0.001), peritoneum (15.4% vs. 28.6%, *P* < 0.001), and bone (20.2% vs. 5.4%, *P* < 0.001) metastases. As for the treatment regimen and lines, there were significant differences in the frequency of Neo*RAS* between testing immediately prior to second-to-fourth-line treatment and to later-line treatment (21.8% vs. 10.7%, *P* = 0.007), prior use and no prior use of regorafenib (6.8% vs. 20.3%, *P* < 0.001), and history and no history of vascular endothelial growth factor inhibitor use (22.5% vs. 11.1%, *P* < 0.001). In Group B, there were significant differences in the frequency of Neo*RAS* between single-organ and multi-organ metastases (15.7% vs. 6.6%, *P* = 0.004), absence and presence of liver metastasis (16.6% vs. 5.6%, *P* < 0.001), and history and no history of vascular endothelial growth factor inhibitor use (11.9% vs 5.2%, *P* = 0.035).

We compared the clinical characteristics of tissue *RAS* WT (*N* = 1077), Neo*RAS* WT (Group A, *N* = 91; Group B, *N* = 42), and *RAS* MT/non-Neo*RAS* WT (*N* = 387) mCRC patients using the GOZILA database (Supplementary Data 2). Neo*RAS* WT mCRC patients had clinically similar characteristics to *RAS* MT mCRC patients. Regarding the primary site, Neo*RAS* WT patients had a higher incidence on the left side colon than *RAS* MT/non-Neo*RAS* WT patients, and the incidence was closer to that in *RAS* WT patients. There was no difference in incidence when the cutoff values for MAF were set at 0.34% and 1% compared with no cutoff.

**Incidence of Neo*RAS* WT according to the tissue *RAS* variants**
The identified *RAS* MTs are listed in Fig. 3. In Group A (*N* = 478), mutations in *KRAS* codons 12 and 13 were detected in 71.3% and 16.3% of patients. Mutations in *KRAS* codon 61 (3.3%), *KRAS* codon 117 (0.8%), *KRAS* codon 146 (3.6%), *NRAS* codon 12 (1.7%), *NRAS* codon 13 (0.6%), and *NRAS* codon 61 (2.3%) were less common (<10% of patients) than mutations in *KRAS* codons 12 and 13. The incidences of Neo*RAS* WT in *KRAS* codon 12, *KRAS* codon 13, *KRAS* codon 61, *KRAS* codon 146, *NRAS* codon 12, and *NRAS* codon 61 were 18.5%, 15.4%,

31.3%, 35.3%, 37.5%, and 18.2%, respectively. In Group A, the frequency of Neo*RAS* WT in *KRAS* exons 3 and 4 MT or *NRAS* MT mCRC tended to be higher than that in *KRAS* exons 2 and 3 MT (27.1% vs. 17.9%, *P* = 0.11)

In Group B (*N* = 429), mutations in *KRAS* codons 12 and 13 were detected in 70.9% and 17.0% of patients. Mutations in *KRAS* codon 61 (3.0%), *KRAS* codon 117 (0.9%), *KRAS* codon 146 (3.5%), *NRAS* codon 12 (1.4%), *NRAS* codon 13 (0.7%), and *NRAS* codon 61 (2.6%) were less common (<10% of patients) than mutations in *KRAS* codons 12 and 13. The incidences of Neo*RAS* WT in *KRAS* codon 12, *KRAS* codon 13, *KRAS* codon 61, *KRAS* codon 146, *NRAS* codon 12, and *NRAS* codon 61 were 8.6%, 9.6%, 15.4%, 26.7%, 16.7%, and 18.2%, respectively. The frequency of Neo*RAS* WT in *KRAS* exons 3 and 4 MT or *NRAS* MT mCRC also tended to be higher than that in *KRAS* exons 2 and 3 MT in Group B (17.3% vs. 8.8%, *P* = 0.076).

**Multivariate analysis results**
In the multivariate logistic regression analysis in Group A, absence of liver (odds ratio [OR]: 0.18, 95% confidence interval [CI]: 0.09−0.34, *P* < 0.00001) and lymph node (OR: 0.46, 95% CI: 0.22−0.94, *P* = 0.033) metastases was significantly related to the presence of Neo*RAS* WT mCRC (Fig. 4a). Similarly, in Group B, absence of liver metastasis and tissue *RAS* MTs other than *KRAS* exon 2 MTs were significantly related to the presence of Neo*RAS* WT mCRC (liver metastasis: OR: 0.29, 95% CI: 0.15−0.59, *P* = 0.0005; original *RAS* status: OR: 2.33; 95% CI: 1.0−5.42, *P* = 0.049; Fig. 4b).

**Clinical outcomes of Neo*RAS* WT mCRC patients treated with anti-EGFR therapy**
In this cohort, six Neo*RAS* WT patients were treated with anti-EGFR mAb-based therapy regimens. The treatment lines ranged from fourth to seventh, and the treatment regimens included anti-EGFR mAb monotherapy (*N* = 2) and mAb in combination with chemotherapy (*N* = 4). Among these 6 patients, 1 patient had partial response (PR), and another 2 patients had stable disease (SD) for at least 6 months. A summary of the treatment courses of the six patients is presented in Fig. 5.

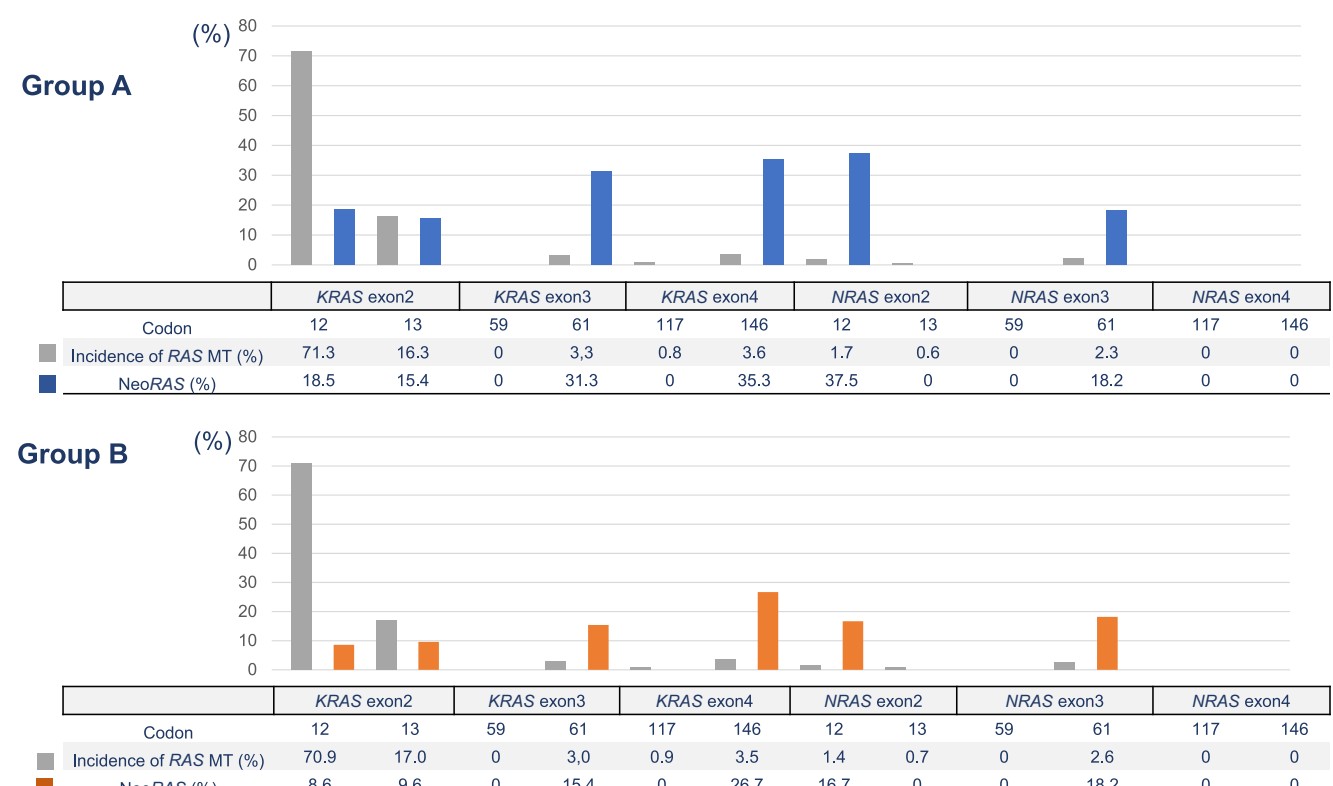

**Fig. 3 | Incidence of tissue *RAS* MT and Neo*RAS* WT according to the tissue *RAS* variants.** The identified *RAS* MTs are listed. In both Group A and Group B, the frequency of Neo*RAS* WT in *KRAS* exons 3 and 4 MT or *NRAS* MT mCRC tends to be higher than that in *KRAS* exons 2 and 3 MT. Source data are provided as a Source Data file. *RAS* rat sarcoma viral oncogene homolog, MT mutant, mCRC metastatic colorectal cancer, WT wild-type.

## Representative cases of Neo*RAS* WT mCRC

We describe below two representative cases of mCRC patients undergoing standard-of-care regimens for *RAS* MT mCRC and reverting from *RAS* MT to Neo*RAS* WT.

Case 1 was a 57-year-old woman with metachronous hepatic and lung-metastasized sigmoid cancer (Fig. 6) Initially, this patient was diagnosed with localized sigmoid colon adenocarcinoma. Primary resection was performed followed by adjuvant chemotherapy. The primary tumor and lymph node lesions demonstrated the presence of *NRAS* G12C MT, although the MAF was close to the test's cutoff value (primary average MAF: 5.0%, lymph node MAF: 1%) according to whole-exome tissue sequencing (Supplementary Table 1). On genomic profiling of tissue samples, using both whole-exome sequencing and FoundationOne CDx®, *NRAS* G12C MT, *APC* MT (R1114*, K1310fa), and *TP53* MT (R175H) were detected. Shortly after adjuvant chemotherapy, liver and lung metastases were detected. Genomic profiling by plasma NGS (Guardant 360®) before chemotherapy initiation revealed *APC* MT (R1114*, K1310fa) and *TP53* MT (R175H), but *NRAS* G12C MT was not detected.

Case 2 was a 46-year-old woman with synchronous hepatic-, lung-, and ovary-metastasized rectal cancer (Fig. 7). Primary resection and metastasectomy of the ovary were performed before chemotherapy initiation because of rectal stenosis. On genomic profiling using whole-exome sequencing, an *NRAS* MT (Q61H) in both primary (average MAF: 55.7%) and metastatic (average MAF: lymph node, 13.7%; ovary, 33.2%) tumors was detected (Supplementary Table 2). This patient had received a first-line standard-of-care regimen (capecitabine, oxaliplatin and bevacizumab [BV]) that led to PR. Therefore, surgery for liver metastasis was performed. *NRAS* MT (Q61H) was detected, but MAF was very low at 0.95%, and relative clonality of *NRAS* was also low at 26.8% (Supplementary Table 2). The number of lung metastases increased after liver metastasectomy, and radical resection was

considered difficult; hence, we decided to continue chemotherapy. The chemotherapy regimen was switched to folinic acid, 5-fluorouracil, and irinotecan (FOLFIRI) + BV owing to chemotherapy-induced peripheral neuropathy. After 1 year and 6 months of chemotherapy, the number of lung metastases did not increase and continued to decrease, and the patient underwent lung metastasectomy. Repeat whole-exome sequencing was performed using a lung metastasis specimen, and *NRAS* MT (Q61H) was detected (average MAF: 33.4%) (Supplementary Table 3). Approximately 1 year after lung metastasectomy, peritoneal metastases occurred, and FOLFIRI + BV was restarted. After disease progression following FOLFIRI + BV treatment, alterations including *NRAS* Q61H were not detected using plasma NGS (Guardant 360®).

Except the two cases for which exome sequencing was performed, there were 18 cases for which tissue NGS data (SCRUM-JAPAN) were available. Supplementary Data 3 shows a comparison between tissue and ctDNA genomic data.

## Discussion

Patients with *RAS* MT mCRC who have failed chemotherapy have a poor prognosis and few effective treatment options. In the current study, we defined Neo*RAS* WT as a meaningful subset of *RAS* MT mCRC, with a prevalence of nearly 10%, that might benefit from available anti-EGFR mAb-based therapies. Previous studies in this area reported highly variable results[17,21–24,30,31] (Supplementary Table 3) owing, in varying degrees, to small sample sizes, divergent definitions of the indication, and limited diagnostic methods. Critically, previous reports examining the incidence of Neo*RAS* WT mCRC have measured only *RAS* MTs to determine the *RAS* mutational status. For example, Sato et al. reported conversion from *RAS* MT to *RAS* WT after first-line chemotherapy in 27 out of 62 (43.5%) patients, with a higher incidence in patients with lung metastases only, those who underwent primary

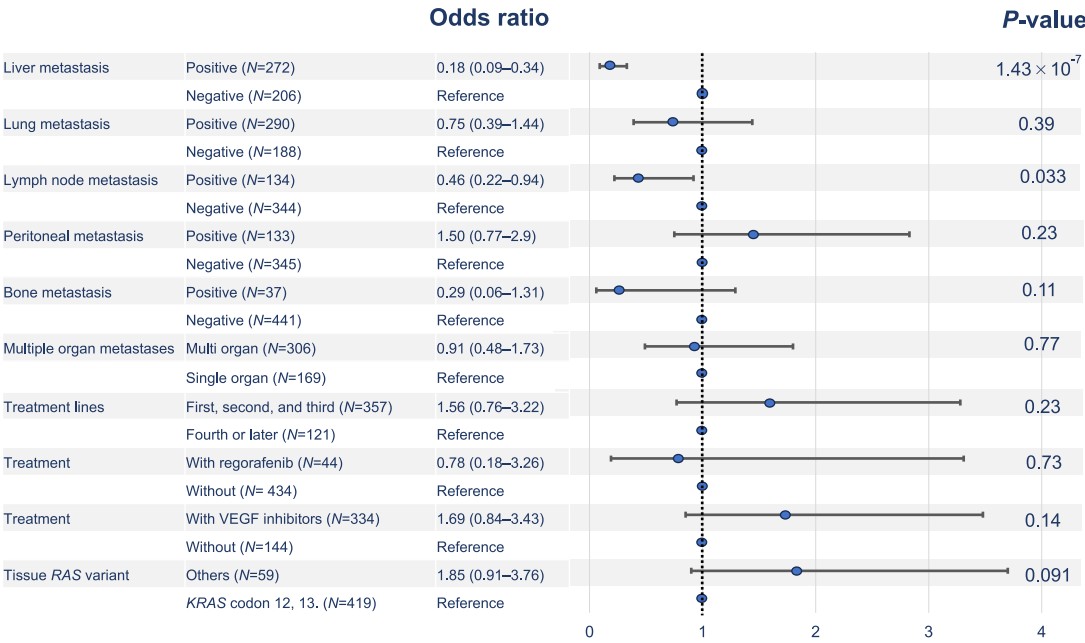

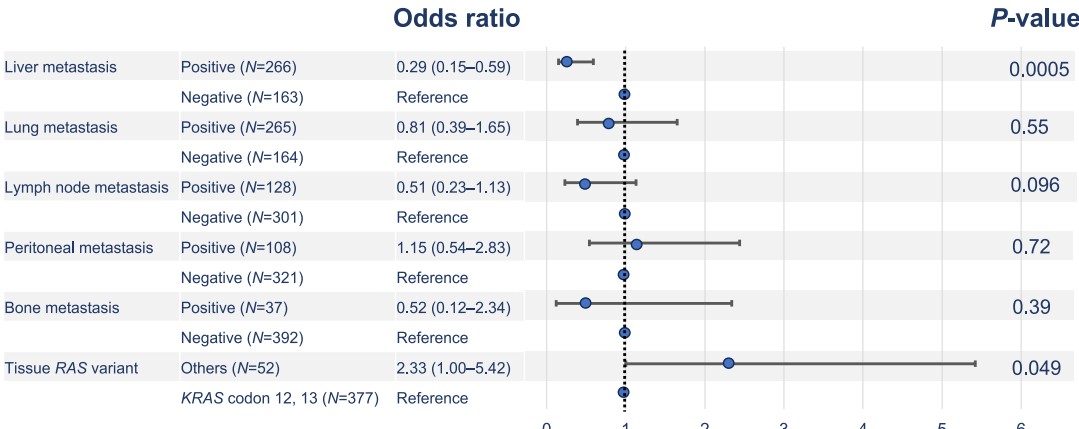

**Fig. 4 | Multivariate analysis results.** Forest plot depicting the logistic regression multivariate analysis for the appearance of Neo*RAS* WT mCRC in Group A (4a) and Group B (4b). The adjusted odds ratio (OR: blue plot) and 95% confidence interval (CI: black horizontal lines) are shown for each factor. Various clinical factors and their association with the appearance of Neo*RAS* WT mCRC, as indicated by OR, are analyzed across the cohort using the two-sided Wald chi-squared test. The unadjusted OR of the appearance of Neo*RAS* WT mCRC is calculated as X/Y. For instance, about liver metastasis, X is defined as "the proportion of mCRC with liver metastases that became Neo*RAS* WT mCRC," and Y is defined as "the proportion of mCRC without liver metastases that became Neo*RAS* WT mCRC." An adjusted OR is an OR that has been adjusted to account for other predictor variables in the model (adjusted OR: 0.18, 95% CI: 0.09–0.34, $P = 1.43 \times 10^{-7}$). Vertical dotted line: the null hypothesis. Number of events: Group A = 91; Group B = 42. Source data are provided as a Source Data file. mCRC metastatic colorectal cancer, WT wild-type, *RAS* rat sarcoma viral oncogene homolog, VEGF vascular endothelial growth factor.

tumor resection, and those who achieved first-line treatment response[30]. In addition, Sunakawa et al. reported that *RAS* MTs were undetected during disease progression in 62% of the patients based on the ctDNA data of 29 patients who were treated with modified FOL-FOXIRI + BV as first-line chemotherapy (JACCRO CC-11 trial)[24]. However, the measurement of *RAS* MTs is influenced by the characteristics of the ctDNA assay, and this may lead to a high false-negative rate, especially when the tumor volume is low. Moreover, some ctDNA assays use cutoff values rather than limit of detection of ctDNA to determine wild-type and mutant clones; thus, even if a mutant clone is detected, it may still be reported as wild-type[31].

To overcome this limitation, the presence of ctDNA should be confirmed using NGS or profiling of methylated genes. However, when methylated genes are used as the basis for the presence of ctDNA, the difference in sensitivity to detect somatic alteration should be taken into consideration. Henry et al. used NGS to assess the loss of *RAS* MT in patients with mCRC who had not previously been treated with anti-EGFR mAb. They reported that the conversion rate of *RAS* status ranged from 2% to 8% and that changes in *RAS* MT status from MT to WT in mCRC were relatively rare[21]. In addition, Nicolazzo et al. reported that using either NGS or methylation assay was a reliable method for confirming the presence of ctDNA in patients with Neo*RAS* WT mCRC[25]. The incidence of Neo*RAS* WT was 37.5% in patients with *RAS* MT mCRC diagnosed using both primary tumor tissue and ctDNA prior to treatment and treated with at least first-line chemotherapy[25]. These results suggest that when evaluating the incidence of Neo*RAS* WT mCRC, the results obtained from ctDNA should be interpreted carefully. In addition, the results would be more reliable if MTs other than *RAS*, such as *APC* MTs or methylated genes, could be confirmed.

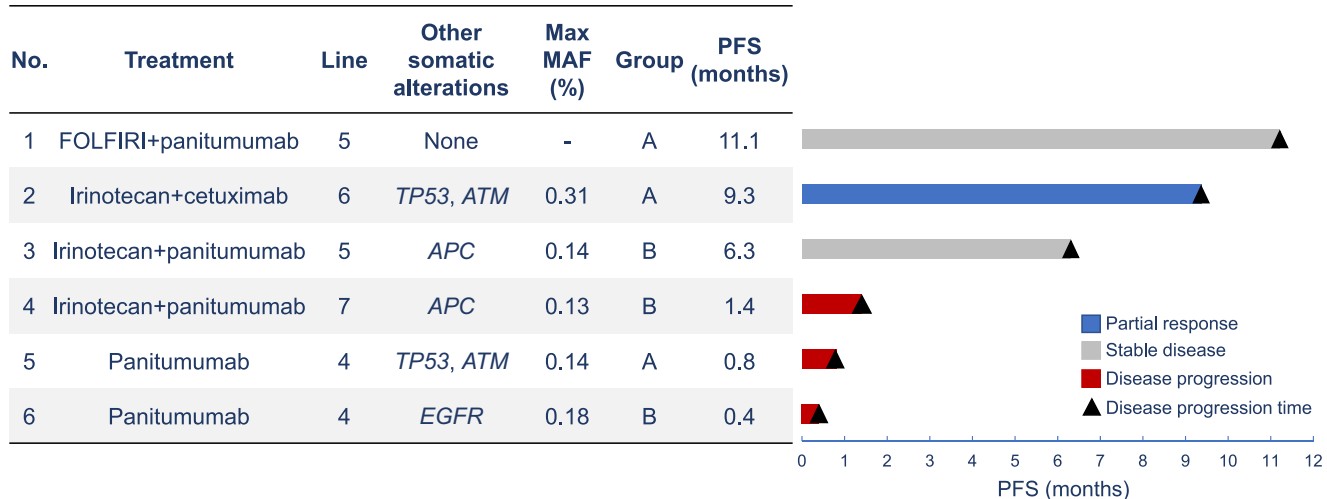

| No. | Treatment | Line | Other somatic alterations | Max MAF (%) | Group | PFS (months) |
|---|---|---|---|---|---|---|
| 1 | FOLFIRI+panitumumab | 5 | None | - | A | 11.1 |
| 2 | Irinotecan+cetuximab | 6 | *TP53, ATM* | 0.31 | A | 9.3 |
| 3 | Irinotecan+panitumumab | 5 | *APC* | 0.14 | B | 6.3 |
| 4 | Irinotecan+panitumumab | 7 | *APC* | 0.13 | B | 1.4 |
| 5 | Panitumumab | 4 | *TP53, ATM* | 0.14 | A | 0.8 |
| 6 | Panitumumab | 4 | *EGFR* | 0.18 | B | 0.4 |

**Fig. 5 | Clinical outcomes of anti-EGFR therapy for patients with Neo*RAS* WT mCRC (*N* = 6).** Among the 6 Neo*RAS* WT patients who are treated with chemotherapy, including anti-EGFR therapy, 1 patient have partial response, and another 2 patients have stable disease for at least 6 months. EGFR: epidermal growth factor receptor, *RAS*: rat sarcoma viral oncogene homolog WT wild-type, FOLFIRI a combination of calcium folinate and fluorouracil with irinotecan hydrochloride hydrate, PFS progression-free survival, MAF mutant allele frequency, NA not available.

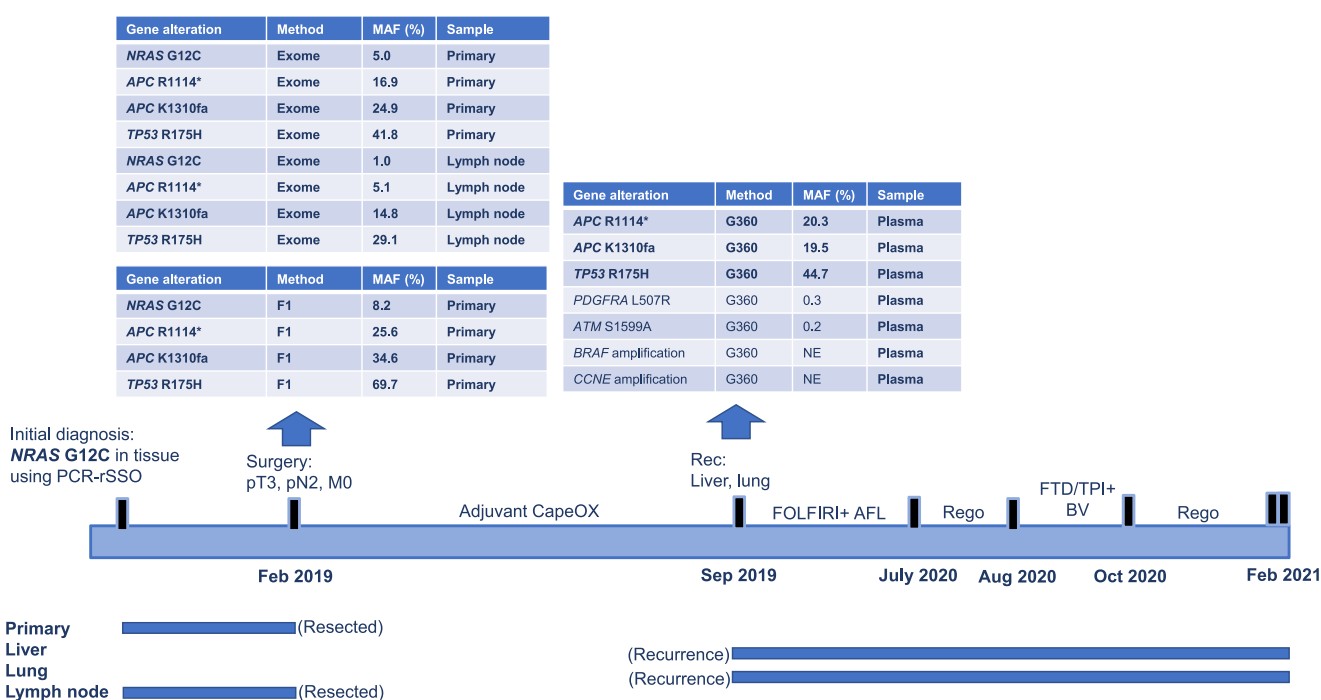

**Fig. 6 | Representative Case 1 of Neo*RAS* WT mCRC.** mCRC metastatic colorectal cancer, WT wild-type, *RAS* rat sarcoma viral oncogene homolog, MAF mutant allele frequency, PCR-rSSO polymerase chain reaction-reverse sequence-specific oligonucleotide, CapeOX combination of capecitabine and oxaliplatin, FOLFIRI combination of calcium folinate, fluorouracil, and irinotecan hydrochloride hydrate, AFL aflibercept, Rego regorafenib, FTD/TPI trifluridine tipiracil hydrochloride, BV bevacizumab.

Although the mechanism of conversion to Neo*RAS* WT mCRC remains unclear, one hypothesis is that mCRC with low-MAF *RAS* mutations may represent evidence of a mixture of *RAS* MT and WT clones within a tumor and that conversion to *RAS* WT after treatment occurs owing to a decrease in the proportion of *RAS* MT clones relative to *RAS* WT[32]. Similarly, Henry et al. reported that the loss of *RAS* MT is associated with a low pretreatment MAF[21]. In this study, the incidence of Neo*RAS* WT in tumors with *KRAS* exon 2 MT status tended to be lower than that in tumors with other *RAS* MTs. Furthermore, multivariate analysis showed that tissue *RAS* MTs other than *KRAS* exon 2

MTs were significantly related to the emergence of Neo*RAS* WT. This finding may reflect an underlying tendency of MTs other than *KRAS* exon 2 MTs toward increased clonal diversity, with both *RAS* MT and WT clones coexisting prior to therapy. This would explain the observed increase in Neo*RAS* WT conversion. Previous reports support our findings. Sato et al. reported that all patients (*N* = 4) with tissue *RAS* Q61H mutations had Neo*RAS* WT[30]. Yoshinami et al. also showed a high incidence of Neo*RAS* among patients with tissue *RAS* MTs in exons other than the *KRAS* exon 2 (5/8, 62.5%), and this finding was related to the emergence of Neo*RAS* WT in their multivariate analysis[33,34]. Loree

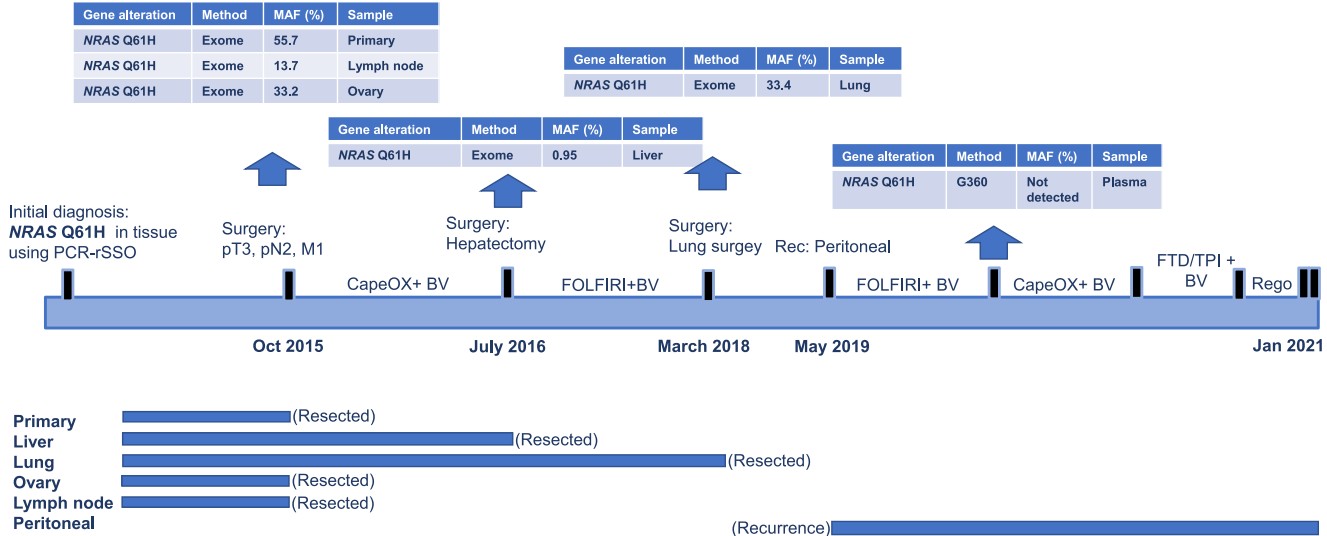

**Fig. 7 | Representative Case 2 of NeoRAS WT mCRC.** mCRC metastatic colorectal cancer, WT wild-type, *RAS* rat sarcoma viral oncogene homolog, MAF mutant allele frequency, PCR-rSSO polymerase chain reaction-reverse sequence-specific oligonucleotide, CapeOX combination of capecitabine and oxaliplatin, BV bevacizumab, FOLFIRI combination of calcium folinate, fluorouracil, and irinotecan hydrochloride hydrate, FTD/TPI trifluridine tipiracil hydrochloride, Rego regorafenib.

et al.[35] similarly described that the allelic fractions of MTs in *KRAS* exons 3 and 4 and *NRAS* were lower than those of *KRAS* exon 2 MTs.

mCRC with low MAF of *RAS* may reflect part of tumor heterogeneity. The two main types of heterogeneity in mCRC are spatial and temporal heterogeneity. Spatial heterogeneity includes inter- or intratumor heterogeneity and inter-metastatic heterogeneity. Regarding metastatic sites, mCRC patients with liver and lymph node metastases had a low frequency of NeoRAS WT in this study. In a previous meta-analysis, the pooled discordance proportion of *KRAS* between the primary tumor and metastatic site was 8% (95% CI: 5–10%)[36], with a relatively high clonality. In addition, there have been several studies on liver metastasis and *KRAS* exon 2 MTs. Therefore, *RAS* clonality may differ among metastatic sites and *RAS* variants[36,37]. In temporal heterogeneity, clonal selection develops, and a series of secondary resistance mechanisms is established during systemic therapies. The representative cases of NeoRAS WT mCRC in this study are shown in Figs. 6 and 7. In Case 1, the genetic alterations were different between the primary tumor and lymph node metastasis, with intratumor and inter-metastatic heterogeneity. Treatment (adjuvant chemotherapy) caused clonal selection, which could have eliminated the *RAS* mutant clones. Therefore, the original tumor had a subclonal *NRAS* MT, which was unlikely to be driving the cancer and was simply lost during treatment.

In Case 2, the gene alterations were also different among the primary tumor, lymph node metastasis, and synchronous ovarian metastasis, with intratumor and inter-metastatic heterogeneity. Treatment (palliative chemotherapy) caused clonal selection and decreased both variant allele frequency and clonality of *RAS*. Therefore, clonal *NRAS* MT, which was outcompeted by a *NRAS* WT subclone, could result in a reversion from *RAS* MT to a phenotypically distinct *RAS* WT. One of the central roles of ctDNA is to overcome the difficulty of obtaining comprehensive genetic information owing to heterogeneity. ctDNA results should be interpreted carefully because ctDNA reflects comprehensive genomic information of the tumor that is easier to detect in mCRC patients with liver and lymph node metastases than in those with lung or peritoneal metastases[38]. To confirm this hypothesis, future analyses should compare the MAF and concordance of *RAS* status between liver and lymph node metastasis and other metastatic sites in patients with *KRAS* exon 2 and in those with

other *RAS* MTs using samples of both tissue and ctDNA before and after treatment.

Furthermore, the NeoRAS WT phenomenon mainly occurs in bevacizumab-treated patients, confirming previous reports that bevacizumab as a first-line therapy is an independent predictor of *RAS* mutation clearance in ctDNA at the time of progressive disease[39]. However, the biological rationale for the association of bevacizumab with *RAS* mutation clearance is currently unknown. The present study observed a significant difference in univariate analysis, but it disappeared in multivariate analysis. Therefore, further research is needed to determine if bevacizumab history increases the frequency of NeoRAS WT.

The major concern related to NeoRAS WT mCRC is whether anti-EGFR mAbs have chemotherapeutic effects. In the present study, three of the six patients with NeoRAS WT mCRC treated with anti-EGFR mAb as a later-line treatment showed treatment response or long progression-free survival (PFS). Before considering the efficacy of anti-EGFR mAbs against NeoRAS WT mCRC, the clinical outcomes of anti-EGFR mAbs against low-MAF *RAS* mCRC should be reviewed. In the ad-hoc biomarker analysis of the CO.17 trial, 1 patient with a *KRAS* A59T MT (MAF = 2%) responded to cetuximab[35]. Furthermore, in the post-hoc analysis of the CRYSTAL trial, patients with low extended *RAS* MT levels, with signals between 0.1% and 5%, benefited from the addition of cetuximab. Similar findings were reported by Laurent-Puig et al., who found that patients with 1% mutated *KRAS* alleles had a similar response to patients with *KRAS* WT[40], suggesting a potential gradient of efficacy based on the MAF of mutant *RAS* in the tumor[6]. The status of low-*RAS* MAF mCRC may change from MT to WT after treatment[32]. Therefore, anti-EGFR mAb is expected to have a chemotherapeutic effect on NeoRAS WT mCRC.

Several prospective and retrospective studies on the chemotherapeutic effects of anti-EGFR mAbs on NeoRAS WT mCRC have been conducted[16–18,22,30,41]. Mohamed et al. conducted a proof-of-concept study of anti-EGFR mAbs in patients with NeoRAS WT mCRC. The main results were an objective response rate of 55.6% and a PFS of 9 months in patients with Neo *RAS* WT mCRC who were treated with a combination of FOLFIRI and cetuximab[41]. Most patients received two or more chemotherapy regimens prior to FOLFIRI and cetuximab[41]. In addition, Nicolazzo et al. reported the efficacy of anti-EGFR mAb in combination with standard chemotherapy for NeoRAS WT mCRC

identified by NGS and/or methylation analysis when patients were resistant to first-line therapy[25]. Among the 20 patients enrolled at the time of first-line treatment resistance, 6 patients received anti-EGFR mAbs and had a median PFS of 6.5 months[25]. Among the 20 patients enrolled at the time of second/third-line treatment resistance, 4 patients received anti-EGFR mAbs and had a median PFS of 6.0 months[25]. This result suggests that anti-EGFR mAbs may be effective for patients with NeoRAS WT mCRC. Therefore, in clinical practice, the RAS status of mCRC patients with tissue RAS MT other than in KRAS exon 2 and without liver or lymph node metastases should be re-assessed using ctDNA at the time of progressive disease, and the indication for treatment with anti-EGFR mAb should be explored. Several clinical trials are ongoing to evaluate the therapeutic efficacy of anti-EGFR mAbs against NeoRAS WT mCRC[19,20].

Our study has some limitations. It lacked information on ctDNA prior to treatment. Furthermore, the number of EGFR antibody-treated individuals was too small to make definitive conclusions. In addition, details of genetic alterations in pretreatment tissues of NeoRAS WT patients who responded to treatment with anti-EGFR mAb were unclear.

Moreover, there was no consensus on the definition of CH in mCRC. In this study, CH was identified only on a reported basis, and DNA from white blood cells was not examined. Finally, the sensitivity of ctDNA detection may have affected the study results. We did not set a detection limit for MAF, and thus, the incidence of NeoRAS WT in Group B was defined as the absence of detectable RAS MT, but detectable for other alterations regardless of MAF, excluding possible CH alterations. As reported in a previous study, Gardant 360 had predictive sensitivities of >98%, 84.0%, and 50% when MAF was set as ≥1%, >0.34%–<1%, and ≤0.34%, respectively[42]. Future technology that can accurately determine the presence of ctDNA with a cutoff neighborhood is desirable, and further large-scale analyses using both tissue and ctDNA information before and after treatment are needed to accurately clarify the incidence and clinicopathological characteristics of NeoRAS WT mCRC.

In conclusion, NeoRAS WT mCRC is associated with the absence of liver and lymph node metastases, as well as RAS MTs other than KRAS exon 2 MTs. Therefore, treatment with anti-EGFR mAbs may be effective in patients with NeoRAS WT mCRC.

## Methods

### Patients

This study was approved by the Institutional Review Board of the Cancer Institute Hospital of the Japanese Foundation of Cancer Research (IRB number: 2021-GB-009) and was conducted in accordance with the principles of the Declaration of Helsinki. The protocol was described on the hospital's website, and the participants were given the opportunity to opt out of the study.

NGS data (Guardant360®, Guardant Health, Inc, Redwood City, CA, USA) were used to confirm the incidence of NeoRAS WT mCRC. Patients with mCRC enrolled in the GOZILA study between March 2018 and February 2022 and who exhibited tissue RAS MTs (KRAS or NRAS exon 2, 3, and 4 mutations) before chemotherapy initiation were eligible, regardless of the treatment line. Briefly, the GOZILA study is a nationwide plasma genomic profiling study involving 31 core cancer institutions in Japan. Patients with metastatic gastrointestinal cancers are eligible for enrollment[26].

### Blood samples, circulating tumor DNA isolation, and sequencing

NGS of ctDNA was performed using the Guardant360® system at Guardant Health. Guardant360 detects single-nucleotide variants, indels, fusions, copy number alterations, and microsatellite instability in 74 genes[26]. In the GOZILA study, $2 \times 10$ mL whole blood was collected in a cell-free DNA BCT® (Streck, Inc., La Vista, NE, USA) and sent to Guardant Health. Then, 5–30 ng of cell-free DNA (cfDNA) isolated

from plasma was labeled with non-redundant oligonucleotides ("molecular barcoding"), enriched using targeted hybridization capture, and sequenced on the NextSeq 550 platform (Illumina, San Diego, CA, USA)[26]. Base call files generated using RTA software version 2.12 (Illumina, San Diego, CA, USA) were demultiplexed using bcl2fastq (version 2.19) and processed using a custom pipeline for molecular barcode detection, sequencing adapter trimming, and base quality trimming. Processed reads were aligned to hg19 using the Burrows–Wheeler aligner-MEM algorithm (arXiv:1303.3997v2). The ctDNA fraction was determined using the maximum variant allelic fraction[26].

### Tumor tissue DNA sequencing

All patients were diagnosed with mCRC based on the analysis of tissue RAS MTs. The mutation profiles of the tissue samples were determined using standard-of-care procedures validated by each hospital. For example, the RASKET-B KIT (MBL, Nagoya, Japan), which applied the PCR-reverse sequence-specific oligonucleotide (PCR-rSSO) method, was used according to the manufacturer's protocol[43]. The PCR-rSSO and Luminex MAP technologies allow multiplex molecular testing in a single well. We examined 12 types of RAS exon 2 mutations (G12S, G12C, G12R, G12D, G12V, G12A, G13S, G13C, G13R, G13D, G13V, and G13A), 8 types of RAS exon 3 mutations (A59T, A59G, Q61K, Q61E, Q61L, Q61P, Q61R, and Q61H), and 4 types of RAS exon 4 mutations (K117N, A146T, A146P, and A146V), as previously described[43]. The procedure of whole-exome sequencing is shown in the Supplementary Note[44,45]. Relative clonality was defined as "subclonal" if the MAF was less than 30% of the highest MAF in the sample and was defined as "clonal" if it was above this threshold[29].

### Endpoints

The primary endpoint was confirmation of the incidence of NeoRAS WT mCRC using NGS. The patients were divided into two groups based on genetic alterations, as follows: Group A, with tissue-confirmed pretreatment RAS MTs and no post-treatment RAS MTs detected on ctDNA analysis; Group B, with tissue-confirmed pretreatment RAS MTs, no post-treatment RAS MTs detected on ctDNA analysis, and other gene MTs confirmed post-treatment without CH. The following genes reported to be potentially CH genes in CRC patients[46,47] were excluded: DNMT3A, TET2, TP53, CEBPA, ETV6, HRAS, PDGFRA, KMT2A, EZH2, GATA2, GNAS, PPM1D, ASLX1, SF3B1, SRSF2, CHEK2, and KMT2D. The secondary endpoints were differences in the incidence of NeoRAS WT mCRC according to the clinicopathological factors and chemotherapeutic efficacy of anti-EGFR mAbs against NeoRAS WT mCRC. Whole-exome multiregion spatial sequencing was also performed on DNA to confirm intratumor heterogeneity in representative NeoRAS WT mCRC ($N = 2$).

### Data collection

Data on the following clinicopathological factors were collected: age, sex, primary tumor location, metastatic site, timing of sample collection, treatment regimen, treatment lines at the time of sampling, and therapeutic effects depending on the tissue RAS variant. Complete response (CR), PR, SD, and progressive disease were defined following the Response Evaluation Criteria in Solid Tumours guidelines, v1.1[48]. The response rate indicated the proportion of patients who achieved CR or PR to chemotherapy, and the disease control rate indicated the proportion of patients who had a CR, PR, or SD response to therapy. PFS was defined as the time from the first day of anti-EGFR therapy to the first objective evidence of disease progression or death from any cause.

### Statistical analysis

Clinicopathological factors were compared between patients with NeoRAS WT and with RAS MT mCRC. Continuous variables were

compared with the Mann–Whitney $U$ test, and between-group comparisons were performed with the $\chi^2$ test and Fisher's exact test. Univariate and multivariate logistic regression analyses were performed to examine the factors related to Neo*RAS* WT mCRC. All factors that showed significance in the univariate analysis were included in the multivariate analysis. All statistical analyses were performed using the statistical software "EZR" (Saitama Medical Center, Jichi Medical University, Saitama, Japan) in R and R commander[49]. All tests were two-sided, and $P < 0.05$ was considered significant. We have uploaded the raw data used in this study as a Source Data file.

### Reporting summary

Further information on research design is available in the Nature Portfolio Reporting Summary linked to this article.

## Data availability

The authors declare that all variant data used in the conduct of the analyses are available within the article and its supplementary information. The datasets, including individual participant data supporting the results reported in this article, will be made available within 3 months from initial request to researchers depending on methodological considerations. The initial contact for the request will be made with the corresponding authors Eiji Shinozaki (eiji.shinozaki@jfcr.or.jp) and Takayuki Yoshino (tyoshino@east.ncc.go.jp). In this cohort, raw ctDNA sequence data was not provided by Guardant Health, other than the source data used for the analysis because of data privacy regulations and restrictions on their use in patient consent forms. Source data are provided as a source data file. Source data are provided with this paper.

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

## Acknowledgements

The authors would like to thank all the patients and their families, all the investigators, research nurses, and study coordinators who participated in this study. Collaborators with GI-SCREEN and GOZILA study Aichi Cancer Center Hospital: Yukiya Narita and Toshiki Masuishi; Gifu University Hospital: Kazuhiro Yoshida; Kagawa University Hospital: Akihito Tsuji; Kanazawa University: Yuta Adachi and Koushiro Ohtsubo; Kyorin University: Junji Furuse; Kyushu University: Eiji Oki; National Cancer Center Hospital: Ken Kato; Saitama Medical University: Yosuke Horita; Shimane Prefectural Central Hospital: Akiyoshi Kanazawa; Shizuoka Cancer Center: Kentaro Yamazaki; St. Marianna University: Takako Nakajima; University of Tsukuba Hospital: Toshikazu Moriwaki. This work was supported by the National Cancer Center Research and Development Fund (31-A-5 to A. Ohtsu) and SCRUM-Japan Funds (http://www.scrum-japan.ncc.go.jp/index.html).

## Author contributions

Hiroki Osumi (conceptualization: lead; data curation: lead; formal analysis: lead; investigation: lead; methodology: lead; project administration: lead; and writing – original draft: lead). Eiji Shinozaki (conceptualization: equal; project administration: lead; supervision: lead; and writing – review & editing: lead). Yoshiaki Nakamura (funding acquisition: lead; methodology: lead; resources: lead; and visualization: lead; writing – review & editing: lead) Taito Esaki (writing – review & editing: supporting). Hiroya Taniguchi (writing – review & editing: supporting). Hironaga Satake (writing – review & editing: supporting). Yu Sunakawa (writing – review & editing: supporting). Yoshito Komatsu (writing – review & editing: supporting). Yoshinori Kagawa (writing – review & editing: supporting). Tadamichi Denda (writing – review & editing: supporting). Manabu Shiozawa (writing – review & editing: supporting). Taroh Satoh (writing – review & editing: supporting). Tomohiro Nishina (writing – review & editing: supporting). Masahiro Goto (writing – review & editing: supporting). Naoki Takahashi (writing – review & editing: supporting). Takeshi Kato (writing – review & editing: supporting). Hideaki Bando (writing – review & editing: supporting). Kensei Yamaguchi (supervision: supporting; writing – review & editing: supporting). Takayuki Yoshino (funding acquisition: lead; methodology: lead; project administration: lead; resources: lead; supervision: equal; writing – review & editing: supporting).

## Competing interests

Eiji Shinozaki: Takeda Pharma, Merck, Lily, and Chugai Pharma. Yoshiaki Nakamura: Chugai Pharma, Guardant Health AMEA, and Merck Research Funding: Taiho Pharmaceutical (Inst), Guardant Health (Inst), Genomedia (Inst), Chugai Pharma (Inst), Guardant Health (Inst), Seattle Genetics (Inst), and Roche (Inst) Taito Esaki: Lilly, Taiho Pharmaceutical, Daiichi Sankyo, and Chugai Pharma Research Funding: Daiichi Sankyo (Inst), MSD (Inst), Novartis (Inst), Ono Pharmaceutical (Inst), Astellas Pharma (Inst), Lilly (Inst), Bayer (Inst), Nihon Kayaku (Inst), Amgen Astellas Bio-Pharma (Inst), Parexel (Inst), IQVIA (Inst), Quintiles (Inst), Eisai (Inst), Pfizer (Inst), Chugai Pharma (Inst), Syneos Health (Inst), Asahi Kasei Pharma (Inst), Amgen (Inst), Dainippon Sumitomo (Inst), and Dainippon Sumitomo (Inst) Hisateru Yasui: Taiho Pharmaceutical, Chugai Pharma, Bristol-Myers Squibb Japan, Daiichi Sankyo, Lilly, Yakult Honsha, Bayer Yakuhin, Takeda, and Ono Pharmaceutical Research Funding: MSD (Inst), Ono Pharmaceutical (Inst), Astellas Pharma (Inst), AstraZeneca (Inst), and Daiichi Sankyo (Inst) Hiroya Taniguchi: Bayer, Sanofi, Takeda, Chugai Pharma, Taiho Pharmaceutical, Lilly, Merck Serono, Yakult Honsha, Medical & Biological Laboratories Co., Ltd, Bristol-Myers Squibb Japan, MSD K.K, Novartis, Daiichi Sankyo, Mitsubishi Tanabe Pharma, Nippon Kayaku, and Ono Yakuhin Research Funding: Dainippon Sumitomo Pharma (Inst), Array BioPharma (Inst), MSD Oncology (Inst), Ono Pharmaceutical (Inst), Daiichi Sankyo (Inst), Sysmex (Inst), Novartis (Inst), and Takeda (Inst) Satake Hironaga: Bristol-Myers Squibb Co., Ltd., Bayer Co.,

Ltd., Chugai Pharmaceutical Co., Ltd, Daiichi Sankyo Co., Ltd., Eli Lilly Japan Co., Ltd., Merck Bio Pharma Co., Ltd., MSD Co., Ltd., Ono Pharmaceutical Co., Ltd., Sanofi Co., Ltd., Taiho Pharmaceutical Co., Ltd., Takeda Co., Ltd. and Yakult Honsha Co., Ltd. Research funding: Ono Pharmaceutical Co Ltd, Daiichi Sankyo, Taiho Pharmaceutical Co Ltd, and Takeda Pharmaceutical Co., Ltd. Yu Sunakawa: Taiho Pharmaceutical, Chugai Pharma, Takeda, Bayer Yakuhin, Bristol-Myers Squibb Japan, Lilly, Merck, Sysmex, MSD K.K, Ono Pharmaceutical, Daiichi Sankyo, Guardant Health, and Incyte Consulting or Advisory Role: Bristol-Myers Squibb Japan, MSD K.K, Daiichi Sankyo, and Merck Research Funding: Taiho Pharmaceutical, Takeda, Chugai Pharma, Lilly, Sanofi, and Otsuka Yoshito Komatsu: Lilly Japan, Taiho Pharmaceutical, Chugai Pharma, Takeda, Bayer Yakuhin, Bristol-Myers Squibb Co, Sanofi/Aventis, Merck, Yakult Honsha, Ono Pharmaceutical, Nipro Corporation, Moroo Co, Asahi Kasei, Mitsubishi Tanabe Pharma, Otsuka, Medical Review Co., Ltd, and Daiichi Sankyo Research Funding: MSD K.K, Taiho Pharmaceutical, Yakult Honsha, Bayer Yakuhin, DAIICHI SANKYO CO., Ltd, Ono Pharmaceutical, NanoCarrier, Eisai, Sanofi/Aventis, Sysmex, Shionogi, IQVIA, Parexel International Corporation, Astellas Pharma, Mediscience Planning, Sumitomo Dainippon Pharma Co., Ltd, A2 Healthcare, Incyte, Lilly (Inst), Nipro Corporation (Inst), and BeiGene (Inst) Yoshinori Kagawa. Consulting or Advisory Role: Taiho Pharmaceutical, Merck, and Lilly Speakers' Bureau: Lilly, Sanofi, Takeda, Merck, Taiho Pharmaceutical, MSD, Chugai Pharma, Yakult Pharmaceutical, Bayer, and Ono Pharmaceutical. Research Funding: Ono Pharmaceutical Tadamichi Denda: Daiichi Sankyo and Ono Pharmaceutical, Sysmex. Research Funding: Ono Pharmaceutical (Inst), Amgen (Inst), MSD, Pfizer, and Bristol-Myers Squibb Foundation. Shiozawa Manabu: Lilly, Takeda, Taiho, Ono, Yakult, and Merck. Taroh Satoh: Chugai Pharma, Merck Serono, Bristol-Myers Squibb, Takeda, Yakult Honsha, Lilly, Bayer Yakuhin, Ono Pharmaceutical, Merck, Astellas Pharma, Taiho Pharmaceutical, Nihon Kayaku, and Daiichi Sankyo Consulting or Advisory Role: Bayer Yakuhin, Lilly, Ono Pharmaceutical, Takara Bio, Merck Serono, and Nihon Kayaku Research Funding: Yakult Honsha (Inst), Chugai Pharma (Inst), Ono Pharmaceutical (Inst), Sanofi (Inst), Lilly (Inst), Daiichi Sankyo (Inst), Merck (Inst), Merck Serono (Inst), Gilead Sciences (Inst), Dainippon Sumitomo Pharma (Inst), and IQVIA (Inst) Tomohiro Nishina: Taiho Pharmaceutical, Ono Pharmaceutical, Bristol-Myers Squibb Japan, Daiichi Sankyo, Chugai Pharma Research Funding: MSD (Inst), Ono Pharmaceutical (Inst), Astellas Pharma (Inst), Daiichi Sankyo/UCB Japan (Inst), Bristol-Myers Squibb Japan (Inst), Chugai Pharma (Inst), Taiho Pharmaceutical (Inst), and AstraZeneca (Inst). Masahiro Goto: Daiichi Sankyo Company, Limited, Ono Pharmaceutical, Taiho Pharmaceutical, MSD K.K, Takeda, Sumitomo Dainippon Pharma Co., Ltd, Yakult Pharmaceutical, and Lilly Research Funding: Chugai Pharma, Taiho Pharmaceutical, and Nippon Kayaku Naoki Takahashi: Ono Pharmaceutical, Bristol-Myers Squibb Japan, and Taiho Pharmaceutical Takeshi Kato: Chugai Pharma, Ono Pharmaceutical, Takeda, Lilly, and Asahi Kasei Research Funding: Chugai Pharma Hideaki Bando: Taiho, Lilly, and Ono. M.W. Nihon Medi-

Physics. Research grant: Ono Pharmaceutical. Kensei Yamaguchi. Consulting or Advisory Role: Bristol-Myers Squibb Japan, and Daiichi Sankyo. Speakers' Bureau: Chugai Pharma, Bristol-Myers Squibb Japan, Takeda, Taiho Pharmaceutical, Lilly, Ono Pharmaceutical, Daiichi Sankyo, and Merck Research Funding: Ono Pharmaceutical (Inst), Taiho Pharmaceutical (Inst), Daiichi Sankyo (Inst), Lilly (Inst), Gilead Sciences (Inst), Yakult Honsha (Inst), Chugai Pharma (Inst), Boehringer Ingelheim (Inst), Eisai (Inst), MSD Oncology (Inst), Sanofi (Inst), and Bristol-Myers Squibb (Inst) Takayuki Yoshino: Chugai Pharma, Takeda Pharma, Merck, Bayer Yakuhin, Ono Pharmaceutical, and MSD K.K. Consulting Fee: Sumitomo Corp. Research Funding: MSD (Inst), Daiichi Sankyo Company, Limited (Inst), Ono Pharmaceutical (Inst), Taiho Pharmaceutical (Inst), Amgen (Inst), Sanofi (Inst), Pfizer (Inst), Genomedia (Inst), Sysmex (Inst), Nippon Boehringer Ingelheim (Inst), Chugai Pharma (Inst), and Eisai (Inst). No other potential conflicts of interest were reported.
