## [Peer Review File · Nature Communications]

Clinical features associated with NeoRAS wild-type metastatic colorectal cancer: A SCRUM-Japan GOZILA substudyREVIEWER COMMENTS

Reviewer #1 (Remarks to the Author): clinical expertise in metastatic colorectal cancer genomics

The manuscript discusses the data on neo-RAS wild-type and clinical impact of the change of the RAS status in liquid biopsy cases. To my knowledge, this is the largest prospectively collected database on this issue, so far.

The manuscript is well written and the English language is flawless.

The authors should emphasize more on the clinical impact of their findings and should include a short section on it: In cases without hepatic and/ or lymphnode metastases with RAS mutations other than KRAS exon 2 it may be indicated to measure RAS mutations at the time of progression to obtain additional treatment options for the individual patient.

Furthermore, the authors should add a section in either the introduction or discussion explaining the molecular hypothesis of NEO-RASwt mCRC

Reviewer #2 (Remarks to the Author): expertise in ctDNA bioinformatics and evolution

The manuscript by Osumi et al. describes an analysis into the clinical and pathological significance of undetectable RAS mutations in post-treatment plasma ctDNA from patients with RAS-mutant metastatic colorectal cancer (mCRC). Undetected RAS mutation is referred to as "NeoRAS WT" and may reflect tumor clones' reversion from RAS-mutant to RAS-wildtype in response to systemic chemotherapy. Such a reversion may potentially create a therapeutic window in which a formerly RAS-mutant mCRC patient might benefit from anti-EGFR therapy (RAS mutations are a known mechanism for resistance to anti-EGFR therapies in CRC). This is a clinically relevant topic, and moreover the phenomenon of Ras-mutant tumors reverting to Ras-wildtype may potentially expose interesting biology about how dependence on driver mutations changes during cancer evolution. To study the characteristics of NeoRAS WT status, the authors explored a cohort of 478 mCRC patients from their previously published SCRUM-GOZILA trial. These patients had RAS mutations confirmed at diagnosis from tumor tissue, and subsequently had systemic therapy followed by plasma ctDNA genotyping after treatment. If the former RAS mutation was not detected in post-treatment plasma ctDNA, the patient was classified as "NeoRAS WT". The authors go on to correlate NeoRAS WT incidence with clinical and pathological variables, finding that NeoRAS WT incidence correlates with the absence of metastases in several tissues, and varies somewhat based on the original RAS mutation's gene (KRAS or NRAS) and amino acid residue. They then describe responses to anti-EGFR monoclonal antibody therapy in six patients with NeoRAS WT and find partial response or stable disease in three patients.

Unfortunately, the data presented in this study is insufficient to say whether "NeoRAS WT" actually reflects tumor clones' reversion to RAS WT, rather than just reflecting less tumor DNA shedding into circulation. This issue is largely because the authors did not take adequate measures to control for false-negatives (i.e., failure to detect an existing RAS mutation leading to a "NeoRAS WT" classification). Classifying low-shedding patients as NeoRAS WT likely explains the authors' finding that NeoRAS WT patients have fewer liver metastases: smaller, less aggressive tumors (which are known to shed less ctDNA in several cancer types, PMID: 24705333, 23484797) are also less likely to seed metastases. Indeed, the authors show in Table 1 that if no effort is taken to exclude low-shedders (Group A), the correlation between NeoRAS WT and absent liver metastases becomes even stronger. Aside from this major flaw, the statistical analyses in Tables 2 and 3 are nice to see but very superficial. Even if NeoRAS WT reflected true mutant-to-wildtype RAS reversion rather than low ctDNA shedding, the only clearly significant difference they show from RAS mutant mCRC is a lower likelihood to have liver metastases. We are not offered any other data that might yield mechanistic hypotheses or insight into whether NeoRAS WT cancers may be phenotypically distinct from RAS mutant mCRCs. The clinical outcomes for 6 NeoRAS WT are interesting but partial response of a NeoRAS WT mCRC patient to anti-EGFR mABs has already been reported in 2018 (PMID: 35135115), and other insights cannot be generalized from such a small number of patients. In general, this study is very light on scientific content, I believe it should not be published at Nature Communications without substantial revisions.

Major comments:

1. The authors attempted to mitigate the concern of false-negatives explaining their "NeoRAS WT" classification by conducting their analyses in both the overall cohort ("Group A"), as well as in a subset of patients with at least one detectable plasma ctDNA mutation ("Group B"). The problem is that detecting one ctDNA variant is not sufficient evidence that we should expect to see a clonal RAS mutation in these samples. What is the limit of detection (i.e. minimum detectable MAF) for ctDNA-sequencing in these plasma samples, given their coverage and the minimum number of ALT reads required to call variants? The vast majority of the ctDNA mutations found in NeoRAS WT patients seem to have MAFs between 0.08-0.3% (Fig 2). I count 26/42 (62%) of Group B NeoRAS WT patients with only a single non-RAS mutation detected, and they all have MAF<1%. This would be consistent with a low level of ctDNA shedding, such that a sample's largest MAF just barely exceeds the smallest detectable MAF. In these cases, stochastic variation in ctDNA fragments across the genome could easily conceal a clonal RAS mutation (especially if the RAS mutation exists on one chromosome copy, while the detected variant existed on multiple chromosome copies of an amplified genomic region). Perhaps by considering all these factors, the authors could provide some probability estimate that a tumor clone in a NeoRAS WT patient truly reverted to RAS wildtype, and then repeat their analyses in the subset of patients with high-confidence for mutant-to-wildtype reversion.

2. Is there any way to assess whether NeoRAS WT patients clinically or pathologically more closely resemble true RAS-WT compared to RAS-mutant mCRC? Perhaps the authors could mine public datasets or other patients within the GOZILA cohort for RAS-WT and RAS-mutant/non NeoRAS WT mCRC patients to compare their characteristics.

3. Could the authors provide data comparing mutant allele frequencies (or ideally purity/ploidy-corrected cancer cell fractions) for all driver variants in each NeoRAS WT patient's original tumor to the mutant allele frequencies detected in their ctDNA? Currently this information is only provided for two case-reports in Figs S1, S2. However, it is extremely useful for understanding how RAS reversion evolves. For example, Fig S1 suggests a case where the original tumor had a subclonal NRAS G12C which was likely not driving the cancer, which was simply lost during treatment. Conversely, Fig S2 could reflect a clonal NRAS mutation which was outcompeted by a NRAS WT subclone, which could result in a reversion from mutant RAS to a phenotypically distinct wild-type RAS. The first case might provide a better therapeutic opportunity for anti-EGFR mABs, while the second case might be more likely to re-develop a RAS-mutation and thus therapeutic resistance. More examples of the second scenario are particularly interesting, as they may reveal that reversion to wild-type RAS requires a broader shift in driver mutation dependence.

4. There is no data or code provided with this manuscript, so we cannot check exactly what data is used to generate the tables and figures.

Minor comments

1. In several places, the term "NeoRAS" rather than "NeoRAS WT" is used. Is this a typo? Or does this refer to patients without "NeoRAS WT"? Please correct this typo or clarify what "NeoRAS" refers to.

2. Tables 2 and 3 would be more effectively communicated if shown graphically. Table 3 should be shown as a forest plot, and the Ns for each comparison (i.e. Liver met + vs -) should be provided for each variable in the table.

3. Much of the text in the "Incidence of NeoRAS WT according to the tissue RAS variants" section is extremely repetitive and difficult to follow, it would be much more effective to show this information graphically.

4. What is the reference category in Group B "Tissue RAS mutation"?

5. In Fig 3, what does Max VAF% refer to? Is this the VAF for RAS mutations in the initial tumor tissue? Or is this the maximum VAF from ctDNA? If the latter, then why do patients in Group A have mutations listed (I assume the patients listed as "Group A" should have no ctDNA mutations detected).

Reviewer #3 (Remarks to the Author): expertise in RAS mutations in colorectal cancer

The manuscript by Osumi et al. describes patients who experienced the conversion of Ras mutant (MT) metastatic colorectal cancer (mCRC) to Ras wildtype (WT) mCRC following chemotherapy, a phenomenon called "NeoRAS WT" tumor. The authors analyzed circulating tumor DNA (ctDNA) in Japanese patients from the GOZILA study and found that between 10-20% mCRC patients experienced "NeoRAS WT". The authors also reported several cases where patients with "NeoRAS WT" benefitted from subsequent anti-EGFR therapy. Overall, the manuscript is well written, and the data analysis is rigorous. The authors took care to use other mutation markers to confirm that "NeoRAS WT" is not simply due to failure of detecting ctDNA in patient plasma.

A weakness of this paper is a lack of new insight into the mechanism of "NeoRAS WT". In its current form, the paper is mostly descriptive. "NeoRAS WT" has been reported in several previous studies, either anecdotally or in a small cohort of patients. The larger mCRC cohort in this paper provides further support for this observation. A compelling hypothesis of how this happens, as the authors pointed out, is tumor heterogeneity where the Ras MT clone(s) were selectively wiped by chemotherapy whereas Ras WT clones persisted. However, given the relatively low prevalence of "NeoRAS WT" (~10%), it is unclear whether Ras mutation itself is responsible for the tumor cells' sensitivity to chemotherapy, or another mechanism in the Ras WT cells is rendering them resistant to chemotherapy in these cases.

Clinically, an important implication for "NeoRAS WT" is whether these tumors will respond to EGFR-targeted therapy. Previous work and this study have found anecdotal evidence that anti-EGFR therapy can benefit some "NeoRAS WT" patients. This is an important observation. However, the authors did not provide further evidence that may add new insights into this response. For example, for those "NeoRAS WT" tumors that responded to EGFR mAb, did the tumor cells show EGFR amplification/overexpression? Because the authors only looked at ctDNA and did not analyze tumor samples, these mechanistic questions were not addressed. These limitations hamper the novelty of this study in its current form.

RESPONSE TO REVIEWERS' COMMENTS

We sincerely appreciate the reviewers for the valuable and insightful comments on our manuscript. We have revised and improved the manuscript as suggested. Please find below our point-by-point responses to the comments. The revised text in response to the reviewers' comments is highlighted in red.

Reviewer #1 (Remarks to the Author): clinical expertise in metastatic colorectal cancer genomics

The manuscript discusses the data on neo-RAS wild-type and clinical impact of the change of the RAS status in liquid biopsy cases. To my knowledge, this is the largest prospectively collected database on this issue, so far.

The manuscript is well written and the English language is flawless.

The authors should emphasize more on the clinical impact of their findings and should include a short section on it: In cases without hepatic and/ or lymphnoe metastases with RAS mutation s other than KRAS exon 2 it may be inidcated to measure RAS mutations at the time of progression to obtain additional treatment options for the individual patient.

Furthermore, the authors should add a section in either the introduction or discussion explaining the molecular hypothesis of NEO-RASwt mCRC

Response: We thank the reviewer for these very positive comments on our work.

Regarding the hypothesized molecular mechanism of NeoRAS WT mCRC, as already mentioned in the manuscript, we believe that tumor heterogeneity is deeply involved, especially when cancers with low RAS MAF fall below the detection threshold after treatment and change to RAS WT. Furthermore, NeoRAS WT phenomenon occurs mainly in bevacizumab-treated patients, confirming previous report that bevacizumab in first-line therapy is an independent predictor of RAS mutation clearance in ctDNA at PD.¹ However, the biological rationale of why bevacizumab is associated with RAS mutation clearance is currently unknown. In the present study, a significant difference was observed on univariate analysis, but it disappeared on multivariate analysis. Therefore, further research is needed to determine if bevacizumab history increases the frequency of NeoRAS WT.

1. Nicolazzo C, et al. RAS Mutation Conversion in Bevacizumab-Treated Metastatic Colorectal Cancer Patients: A Liquid Biopsy Based Study. *Cancers (Basel)*. **14** (2022).

Accordingly, we have added the following description to the “Discussion” section.

(Page 12, lines 29–33)

Therefore, in clinical practice, the *RAS* status of mCRC patients with tissue *RAS* MT other than *KRAS* exon 2 and without liver or lymph node metastases should be reassessed using ctDNA at the time of progressive disease and the indication for treatment with anti-EGFR mAb should be explored.

(Page 11, lines 26–33)

Furthermore, the Neo*RAS* WT phenomenon mainly occurs in bevacizumab-treated patients, confirming previous reports that bevacizumab as a first-line therapy is an independent predictor of *RAS* mutation clearance in ctDNA at PD.³⁹ However, the biological rationale for the association of bevacizumab with *RAS* mutation clearance is currently unknown. In the present study, a significant difference was observed on univariate analysis, but it disappeared on multivariate analysis. Therefore, further research is needed to determine if bevacizumab history increases the frequency of Neo*RAS* WT.

Reviewer #2 (Remarks to the Author): expertise in ctDNA bioinformatics and evolution

The manuscript by Osumi et al. describes an analysis into the clinical and pathological significance of undetectable *RAS* mutations in post-treatment plasma ctDNA from patients with *RAS*-mutant metastatic colorectal cancer (mCRC). Undetected *RAS* mutation is referred to as “Neo*RAS* WT” and may reflect tumor clones’ reversion from *RAS*-mutant to *RAS*-wildtype in response to systemic chemotherapy. Such a reversion may potentially create a therapeutic window in which a formerly *RAS*-mutant mCRC patient might benefit from anti-EGFR therapy (*RAS* mutations are a known mechanism for resistance to anti-EGFR therapies in CRC). This is a clinically relevant topic, and moreover the phenomenon of *Ras*-mutant tumors reverting to *Ras*-wildtype may potentially expose interesting biology about how dependence on driver mutations changes during cancer evolution. To study the characteristics of Neo*RAS* WT status, the

authors explored a cohort of 478 mCRC patients from their previously published SCRUM-GOZILA trial. These patients had RAS mutations confirmed at diagnosis from tumor tissue, and subsequently had systemic therapy followed by plasma ctDNA genotyping after treatment. If the former RAS mutation was not detected in post-treatment plasma ctDNA, the patient was classified as “NeoRAS WT”. The authors go on to correlate NeoRAS WT incidence with clinical and pathological variables, finding that NeoRAS WT incidence correlates with the absence of metastases in several tissues, and varies somewhat based on the original RAS mutation’s gene (KRAS or NRAS) and amino acid residue. They then describe responses to anti-EGFR monoclonal antibody therapy in six patients with NeoRAS WT and find partial response or stable disease in three patients.

Unfortunately, the data presented in this study is insufficient to say whether “NeoRAS WT” actually reflects tumor clones’ reversion to RAS WT, rather than just reflecting less tumor DNA shedding into circulation. This issue is largely because the authors did not take adequate measures to control for false-negatives (i.e., failure to detect an existing RAS mutation leading to a “NeoRAS WT” classification). Classifying low-shedding patients as NeoRAS WT likely explains the authors’ finding that NeoRAS WT patients have fewer liver metastases: smaller, less aggressive tumors (which are known to shed less ctDNA in several cancer types, PMID: 24705333, 23484797) are also less likely to seed metastases. Indeed, the authors show in Table 1 that if no effort is taken to exclude low-shedders (Group A), the correlation between NeoRAS WT and absent liver metastases becomes even stronger. Aside from this major flaw, the statistical analyses in Tables 2 and 3 are nice to see but very superficial. Even if NeoRAS WT reflected true mutant-to-wildtype RAS reversion rather than low ctDNA shedding, the only clearly significant difference they show from RAS mutant mCRC is a lower likelihood to have liver metastases. We are not offered any other data that might yield mechanistic hypotheses or insight into whether NeoRAS WT cancers may be phenotypically distinct from RAS mutant mCRCs. The clinical outcomes for 6 NeoRAS WT are interesting but partial response of a NeoRAS WT mCRC patient to anti-EGFR mABs has already been reported in 2018 (PMID: 35135115), and other insights cannot be generalized from such a small number of patients. In general, this study is very light on scientific content, I believe it should not be published at Nature Communications without substantial revisions.

Response: We thank the reviewer for the positive comments and valuable feedback.

Major comments:

1. The authors attempted to mitigate the concern of false-negatives explaining their “NeoRAS WT” classification by conducting their analyses in both the overall cohort (“Group A”), as well as in a subset of patients with at least one detectable plasma ctDNA mutation (“Group B”). The problem is that detecting one ctDNA variant is not sufficient evidence that we should expect to see a clonal RAS mutation in these samples. What is the limit of detection (i.e. minimum detectable MAF) for ctDNA-sequencing in these plasma samples, given their coverage and the minimum number of ALT reads required to call variants? The vast majority of the ctDNA mutations found in NeoRAS WT patients seem to have MAFs between 0.08-0.3% (Fig 2). I count 26/42 (62%) of Group B NeoRAS WT patients with only a single non-RAS mutation detected, and they all have MAF<1%. This would be consistent with a low level of ctDNA shedding, such that a sample’s largest MAF just barely exceeds the smallest detectable MAF. In these cases, stochastic variation in ctDNA fragments across the genome could easily conceal a clonal RAS mutation (especially if the RAS mutation exists on one chromosome copy, while the detected variant existed on multiple chromosome copies of an amplified genomic region). Perhaps by considering all these factors, the authors could provide some probability estimate that a tumor clone in a NeoRAS WT patient truly reverted to RAS wildtype, and then repeat their analyses in the subset of patients with high-confidence for mutant-to-wildtype reversion.

Response: Thank you for your comment. In this study, we did not set a detection limit for MAF. Therefore, the incidence of NeoRAS WT in Group B was defined as no detectable RAS MT, but detectable for other alterations regardless of MAF except for possible CH alterations. As reported in a previous study¹, Gardant 360 had predictive sensitivities of >98%, 84.0%, and 50% when MAF was set as $\geq 1\%$, between 0.34 and 1%, and $\leq 0.34\%$, respectively. The incidences of NeoRAS WT mCRC in Group B were 2.5% (10/397) and 1.5% (6/393) with MAF cutoffs of 0.34% and 1%, respectively. Future technology that can accurately determine the cutoff neighborhood are desirable and has been described in the limitations paragraph in the manuscript.

¹. Caughey BA, et al. Identification of an optimal mutant allele frequency to detect activating KRAS, NRAS, and BRAF mutations in a commercial cell-free DNA next-

generation sequencing assay in colorectal and pancreatic adenocarcinomas. *Journal of gastrointestinal oncology*. **14**, 2083-96 (2023).

Accordingly, we have added the following descriptions to the “Results” and “Discussion” sections.

(Page 6, lines 2–5)

When the limits of detection (i.e., minimum detectable mutant allele frequency) were defined as 0.34% and 1%, the incidences of Neo*RAS* WT mCRC in Group B were 2.5% (10/397) and 1.5% (6/393), respectively.

(Page 13, lines 9–15)

In this study, we did not set a detection limit for MAF. Therefore, the incidence of Neo*RAS* WT in Group B was defined as the absence of detectable *RAS* MT, but detectable for other alterations regardless of MAF, excluding possible CH alterations. As reported in a previous study, Gardant 360 had predictive sensitivities of >98%, 84.0%, and 50% when MAF was set as $\geq 1\%$, $>0.34\% < 1\%$, and $\leq 0.34\%$, respectively. Future technology that can accurately determine the presence of ctDNA with a cutoff neighborhood is desirable.

2. Is there any way to assess whether Neo*RAS* WT patients clinically or pathologically more closely resemble true *RAS*-WT compared to *RAS*-mutant mCRC? Perhaps the authors could mine public datasets or other patients within the GOZILA cohort for *RAS*-WT and *RAS*-mutant/non Neo*RAS* WT mCRC patients to compare their characteristics.

Response: Thank you for your comment. We compared the clinical characteristics of *RAS* WT (n=1077), Neo*RAS* WT (Group A n=91, Group B n=42), and *RAS* MT/non-Neo*RAS* WT (n=387) mCRC patients using the GOZILA database. Neo*RAS* WT mCRC was clinically similar to *RAS* MT mCRC. Regarding the primary site, Neo*RAS* WT had a higher incidence on the left side colon compared to *RAS* MT/non-Neo*RAS* WT and seemed closer to *RAS* WT. No change was observed when the cutoff values for MAF were set at 0.34% and 1% compared to no cutoff. We have added these data to Supplemental table S1.

Accordingly, we have added the above findings to the “Results” section.

(Page 6, lines 21–27)

3. Could the authors provide data comparing mutant allele frequencies (or ideally purity/ploidy-corrected cancer cell fractions) for all driver variants in each NeoRAS WT patient's original tumor to the mutant allele frequencies detected in their ctDNA? Currently this information is only provided for two case-reports in Figs S1, S2. However, it is extremely useful for understanding how RAS reversion evolves. For example, Fig S1 suggests a case where the original tumor had a subclonal NRAS G12C which was likely not driving the cancer, which was simply lost during treatment. Conversely, Fig S2 could reflect a clonal NRAS mutation which was outcompeted by a NRAS WT subclone, which could result in a reversion from mutant RAS to a phenotypically distinct wild-type RAS. The first case might provide a better therapeutic opportunity for anti-EGFR mABs, while the second case might be more likely to re-develop a RAS-mutation and thus therapeutic resistance. More examples of the second scenario are particularly interesting, as they may reveal that reversion to wild-type RAS requires a broader shift in driver mutation dependence.

Response: Except for the two cases for which exome sequencing was performed, there were 18 cases for which tissue NGS data (SCRUM-JAPAN) could be used. Supplemental table S4 comparing the tissue and ctDNA genomic data has been provided. Furthermore, since your comments are accurate and insightful, we have changed Figs S1 and 2 to Figs 6 and 7, respectively, and added data relevant to your comments to the manuscript. Thank you very much.

Accordingly, we have added the following details to the "Discussion" section.

(Page 11, lines 9–10)

Therefore, the original tumor had a subclonal *NRAS* MT, which was unlikely to be driving the cancer and was simply lost during treatment.

(Page 11, lines 15–16)

Therefore, clonal *NRAS* MT, which was outcompeted by a *NRAS* WT subclone, could result in a reversion from *RAS* MT to a phenotypically distinct *RAS* WT.

4. There is no data or code provided with this manuscript, so we cannot check exactly what data is used to generate the tables and figures.

Response: Thank you for your comments. We have uploaded the raw data used in this study as Supplemental table 6.

Minor comments

1. In several places, the term “NeoRAS” rather than “NeoRAS WT” is used. Is this a typo? Or does this refer to patients without “NeoRAS WT”? Please correct this typo or clarify what “NeoRAS” refers to.

Response: We thank the reviewer for their careful review. We have corrected the error across the manuscript.

2. Tables 2 and 3 would be more effectively communicated if shown graphically. Table 3 should be shown as a forest plot, and the Ns for each comparison (i.e. Liver met + vs -) should be provided for each variable in the table.

Response: Thank you for your comment. Table 2 has been converted to a bar chart to clearly demonstrate the frequency of each *RAS* variant and the percentage of NeoRAS WT as Fig 3. Table 3 has been converted to a forest plot as Fig 4a and 4b.

3. Much of the text in the “Incidence of NeoRAS WT according to the tissue *RAS* variants” section is extremely repetitive and difficult to follow, it would be much more effective to show this information graphically.

Response: Thank you for pointing this out. Table 2 has been converted to a bar chart to indicate the frequency of each *RAS* variant and the incidence of NeoRAS WT in both Groups A and B.

(Page 6, lines 30–34, Page 7, lines 1–16)

The identified *RAS* MTs are listed in Fig 3. In Group A (n = 478), Mutations in *KRAS* codons 12 and 13 were detected in 71.3% and 16.3% of patients, respectively. Mutations in *KRAS* codon 61 (3.3%), *KRAS* codon 117 (0.8%), *KRAS* codon 146 (3.6%), *NRAS* codon 12 (1.7%), *NRAS* codon 13 (0.6%), and *NRAS* codon 61 (2.3%) were less common (<10% of patients) than mutations in *KRAS* codons 12 and 13.

The incidences of NeoRAS WT in *KRAS* codon 12, *KRAS* codon 13, *KRAS* codon 61, *KRAS* codon 146, *NRAS* codon 12, and *NRAS* codon 61 were 18.5%, 15.4%, 31.3%,

35.3%, 37.5%, and 18.2%, respectively. The frequency of NeoRAS WT in *KRAS* exons 3 and 4 MT or *NRAS* MT mCRC tended to be higher than that in *KRAS* exons 2 and 3 MT in Group A (27.1% vs. 17.9%, $P=0.11$)

In Group B (n = 429), Mutations in *KRAS* codons 12 and 13 were detected in 70.9% and 17.0% of patients, respectively. Mutations in *KRAS* codon 61 (3.0%), *KRAS* codon 117 (0.9%), *KRAS* codon 146 (3.5%), *NRAS* codon 12 (1.4%), *NRAS* codon 13 (0.7%), and *NRAS* codon 61 (2.6%) were less common (<10% of patients) than mutations in *KRAS* codons 12 and 13.

The incidences of NeoRAS WT in *KRAS* codon 12, *KRAS* codon 13, *KRAS* codon 61, *KRAS* codon 146, *NRAS* codon 12, and *NRAS* codon 61 were 8.6%, 9.6%, 15.4%, 26.7%, 16.7%, and 18.2%, respectively. The frequency of NeoRAS WT in *KRAS* exons 3 and 4 MT or *NRAS* MT mCRC also tended to be higher than that in *KRAS* exons 2 and 3 MT in Group B (17.3% vs. 8.8%, $P=0.076$).

4. What is the reference category in Group B “Tissue RAS mutation”?

Response: We thank the reviewer for their careful review. We have corrected the reference category from “Tissue *RAS* mutation” to “Tissue *RAS* variant.”

5. In Fig 3, what does Max VAF% refer to? Is this the VAF for RAS mutations in the initial tumor tissue? Or is this the maximum VAF from ctDNA? If the latter, then why do patients in Group A have mutations listed (I assume the patients listed as “Group A” should have no ctDNA mutations detected).

Response: Thank you for your comment. It shows the MAX MAF for alterations detected in ctDNA, but *TP53* mutations cannot be included in Group B because potentially CH genes in CRC patients were excluded. Therefore, it refers to a case with MAF value in Group A (although it is a reference value).

Reviewer #3 (Remarks to the Author): expertise in RAS mutations in colorectal cancer

The manuscript by Osumi et al. describes patients who experienced the conversion of Ras mutant (MT) metastatic colorectal cancer (mCRC) to Ras wildtype (WT) mCRC following chemotherapy, a phenomenon called “NeoRAS WT” tumor. The authors analyzed circulating tumor DNA (ctDNA) in Japanese patients from the GOZILA study

and found that between 10-20% mCRC patients experienced “NeoRAS WT”. The authors also reported several cases where patients with “NeoRAS WT” benefitted from subsequent anti-EGFR therapy. Overall, the manuscript is well written, and the data analysis is rigorous. The authors took care to use other mutation markers to confirm that “NeoRAS WT” is not simply due to failure of detecting ctDNA in patient plasma.

A weakness of this paper is a lack of new insight into the mechanism of “NeoRAS WT”. In its current form, the paper is mostly descriptive. “NeoRAS WT” has been reported in several previous studies, either anecdotally or in a small cohort of patients. The larger mCRC cohort in this paper provides further support for this observation. A compelling hypothesis of how this happens, as the authors pointed out, is tumor heterogeneity where the Ras MT clone(s) were selectively wiped by chemotherapy whereas Ras WT clones persisted. However, given the relatively low prevalence of “NeoRAS WT” (~10%), it is unclear whether Ras mutation itself is responsible for the tumor cells’ sensitivity to chemotherapy, or another mechanism in the Ras WT cells is rendering them resistant to chemotherapy in these cases.

Clinically, an important implication for “NeoRAS WT” is whether these tumors will respond to EGFR-targeted therapy. Previous work and this study have found anecdotal evidence that anti-EGFR therapy can benefit some “NeoRAS WT” patients. This is an important observation. However, the authors did not provide further evidence that may add new insights into this response. For example, for those “NeoRAS WT” tumors that responded to EGFR mAb, did the tumor cells show EGFR amplification/overexpression? Because the authors only looked at ctDNA and did not analyze tumor samples, these mechanistic questions were not addressed. These limitations hamper the novelty of this study in its current form.

Response: We thank the reviewer for the positive comments and insightful feedback.

Regarding the hypothesis of the molecular mechanism of NeoRAS WT mCRC, which has already been described in the text, we believe that tumor heterogeneity is deeply involved, especially the phenomenon that cancers with low *RAS* MAF fall below the detection limit after treatment and change to *RAS* WT. No research paper mentions heterogeneity as a mechanism of NeoRAS WT mCRC. Furthermore, NeoRAS WT phenomenon occurs mainly in bevacizumab-treated patients, confirming previous reports that bevacizumab as a first-line therapy is an independent predictor of *RAS* mutation clearance in ctDNA at PD. However, the biological rationale why bevacizumab is associated with *RAS* mutation clearance is currently unknown. In the present study, a

significant difference was observed in univariate analysis, but it did not remain a significant factor in multivariate analysis. Therefore, further research is needed to determine if bevacizumab history increases the frequency of Neo*RAS* WT.

Other than the two cases for which exome sequencing was performed, tissue NGS data were available for 18 cases. Unfortunately, we could not identify any cases in which anti-EGFR mAb was used, despite a possibility that EGFR amplification and overexpression may be associated with the therapeutic effect of anti-EGFR mAb in *RAS* WT. Details of genetic alterations in pre-treatment tissues of Neo*RAS* WT patients that responded to treatment with anti-EGFR mAb are not clear. Further research is needed to clarify this point and has been described in the limitations paragraph.

Accordingly, we have added the following description to the “Discussion” section.
(Page 11, lines 26–33)

Furthermore, the Neo*RAS* WT phenomenon mainly occurs in bevacizumab-treated patients, confirming previous reports that bevacizumab as a first-line therapy is an independent predictor of *RAS* mutation clearance in ctDNA at PD. However, the biological rationale for the association of bevacizumab with *RAS* mutation clearance is currently unknown. In the present study, a significant difference was observed on univariate analysis, but it disappeared on multivariate analysis. Therefore, further research is needed to determine if bevacizumab history increases the frequency of Neo*RAS* WT.

(Page 13, lines 3–4)

In addition, details of genetic alterations in pretreatment tissues of Neo*RAS* WT patients who responded to treatment with anti-EGFR mAb are unclear.

REVIEWERS' COMMENTS

Reviewer #1 (Remarks to the Author):

The comments from my side have been sufficiently answered and the manuscript has been revised respectively. With the addition of the sections and the additional literature, the manuscript is now ready to be published.

No more comments from my side

Reviewer #2 (Remarks to the Author):

Review of author point-by-point rebuttal of "Clinical features associated with NeoRAS wild-type metastatic colorectal cancer: A SCRUM-Japan GOZILA substudy":

The authors have adequately addressed my review, I have no further comments.

Reviewer #3 (Remarks to the Author):

The revised manuscript by Osumi et al. provided additional information in two areas: the first is a more detailed discussion on the false-negative rate of detecting mutant Ras allele in ctDNA that could lead to the misclassification of tumors as being "NeoRAS WT". The second is a more detailed description of the clinical and pathological features of "NeoRAS WT" tumors. Although the additional information is helpful, it does not represent a substantial improvement of the study. "NeoRAS WT" tumors following treatment has been described before by multiple studies. Because patient number are small in the "NeoRAS WT" category, it is unclear whether the anecdotal response to EGFR-targeted therapy is generalizable. The lack of new mechanistic insight in this study, as I pointed out in my previous comments, limits the novelty for this study.

RESPONSE TO REVIEWERS' COMMENTS

We sincerely appreciate the reviewers for the valuable and insightful comments on our manuscript. We have revised and improved the manuscript as suggested. Please find below our point-by-point responses to the comments. The revised text in response to the reviewers' comments is highlighted in red in the manuscript.

Reviewer #1 (Remarks to the Author):

The comments from my side have been sufficiently answered and the manuscript has been revised respectively. With the addition of the sections and the additional literature, the manuscript is now ready to be published.

No more comments from my side

Response: We thank Reviewer 1 for the positive comments and valuable feedback.

Reviewer #2 (Remarks to the Author):

Review of author point-by-point rebuttal of "Clinical features associated with NeoRAS wild-type metastatic colorectal cancer: A SCRUM-Japan GOZILA substudy":

The authors have adequately addressed my review, I have no further comments.

Response: We thank Reviewer 2 for the positive comments and valuable feedback.

Reviewer #3 (Remarks to the Author):

The revised manuscript by Osumi et al. provided additional information in two areas: the first is a more detailed discussion on the false-negative rate of detecting mutant Ras allele in ctDNA that could lead to the misclassification of tumors as being "NeoRAS WT". The second is a more detailed description of the clinical and pathological features of "NeoRAS WT" tumors. Although the additional information is helpful, it does not represent a substantial improvement of the study. "NeoRAS WT" tumors following treatment has been described before by multiple studies. Because patient number are small in the "NeoRAS WT" category, it is unclear whether the anecdotal response to

EGFR-targeted therapy is generalizable. The lack of new mechanistic insight in this study, as I pointed out in my previous comments, limits the novelty for this study.

Response: Several studies have investigated Neo*RAS* WT mCRC, but few have addressed the mechanism by which Neo*RAS* WT mCRC develops. To our best knowledge, this is the largest case report to date. Regarding the hypothesis of the molecular mechanism of Neo*RAS* WT mCRC, which has already been described in the text, we believe that tumor heterogeneity is highly involved. We found no research paper that mentions heterogeneity as a mechanism of Neo*RAS* WT mCRC, although further research is needed to clarify other hypothesis of the appearance of Neo*RAS* WT mCRC, as well as the chemotherapeutic effect of anti-EGFR mAb for Neo*RAS* WT mCRC. We thank the reviewer for the comments and valuable feedback.